# Effects of 2-Year Cognitive–Motor Dual-Task Training on Cognitive Function and Motor Ability in Healthy Elderly People: A Pilot Study

**DOI:** 10.3390/brainsci8050086

**Published:** 2018-05-11

**Authors:** Emiko Morita, Hisayo Yokoyama, Daiki Imai, Ryosuke Takeda, Akemi Ota, Eriko Kawai, Yuta Suzuki, Kazunobu Okazaki

**Affiliations:** 1Department of Environmental Physiology for Exercise, Osaka City University Graduate School of Medicine, Osaka 558-8585, Japan; e-morita@pt-u.aino.ac.jp (E.M.); dimai@sports.osaka-cu.ac.jp (D.I.); kawai@respiratorycontrol.com (E.K.); suzuki@sports.osaka-cu.ac.jp (Y.S.); okazaki@sports.osaka-cu.ac.jp (K.O.); 2Department of Physical Therapy Faculty of Health Science, Aino University, Ibaraki 567-0012, Japan; 3Research Center for Urban Health and Sports, Osaka City University, Osaka 558-8585, Japan; wind-05@med.osaka-cu.ac.jp; 4Department of Health and Sports Science, Osaka Electro-communication University, Osaka 572-8530, Japan; ota@osakac.ac.jp

**Keywords:** attention, dual-task training, cognitive function, exercise, prevention

## Abstract

We aimed to examine the effect of 2-year cognitive–motor dual-task (DT) training on cognitive functions and motor ability of healthy elderly people without marked cognitive impairment. From the 25 participants of our 12-week DT trial conducted in 2014, we recruited 8 subjects who voluntarily participated in a new DT training program once a week for 2 years (exercise (EX) group). Their cognitive functions were evaluated by the Modified Mini-Mental State (3MS) examination and the Trail Making Test, and results were compared with those of the 11 subjects who discontinued the training and did not perform any types of exercise for 2 years (non-exercise (NO) group). Subjects in the NO group showed deterioration in the 3MS examination results, especially in the cognitive domain of attention. Meanwhile, participation in DT training maintained the scores in almost all domains of cognitive function, as well as the total 3MS scores. However, both groups had impaired quadriceps muscle strength and motor ability after the 2-year observation period. These results suggest that participating in exercise program comprising DT training for 2 years may be beneficial for maintaining the broad domains of cognitive function in healthy elderly people, although further verification is needed.

## 1. Introduction

According to the World Alzheimer Report 2016, 47 million people suffer from dementia worldwide, and the number of patients with dementia is estimated to reach > 131 million by 2050 [1]. One in five people aged ≥ 65 years have dementia in Japan [2], an unparalleled superaged society. Cognitive impairment not only hampers quality of life [3,4] but causes mental and physical burden to family members and caregivers [5,6]. An increase in the number of patients with cognitive impairment leads to social problems, such as shortage in the number of caregivers and difficulty in securing financial resources at the social level [1,7]. 

To date, many studies have revealed that cognitive training could improve cognitive function in elderly people [8,9,10]. Currently, evidence is also accumulating regarding the effect of exercise as a non-pharmacological therapeutic strategy for the cognitive function of elderly with and without dementia [11,12,13]. We have previously demonstrated that 12-week dual-task (DT) training, which includes simultaneous motor and cognitive tasks, was more effective in improving various domains of cognitive function than single-task training in healthy elderly people [14]. However, most previous investigations, similar to our previous study, comparatively examined the short-term effects of exercise on cognitive function, and how long-term exercise including DT training modulates cognitive function in elderly population remains unclear.

In the present pilot study, we examined the effects of 2-year cognitive–motor DT training on cognitive function and motor ability in healthy elderly people without marked cognitive impairment.

## 2. Materials and Methods

### 2.1. Participants

Subjects were recruited from the participants of our 12-week DT trial conducted in 2014 [14]. At that time, participants were recruited from community dwellers in Sumiyoshi-ku, Osaka City, Japan by advertising in a local magazine. Included patients were healthy, sedentary elderly people, aged over 65 years, with no habit of regular exercise for more than 1 hour per week. Subjects who had a history of ischemic heart disease, chronic heart failure, stroke, severe hypertension, diabetes, marked dementia with a score of under 60 for the Modified Mini-Mental State (3MS) examination, or a neuropsychiatric disorder were excluded from the previous study. One applicant with marked dementia was excluded due to these criteria. Of all 25 participants, eight consecutive subjects voluntarily participated in a new 2-year exercise program from July 2014 to July 2016. The remaining 17 subjects discontinued the training and did not perform any types of exercise for more than 1 h per week for 2 years. All eight subjects who participated in the new program and 11 of the 17 subjects who discontinued the training consented to participate in the present study, and they were assigned to the exercise (EX) and non-exercise (NO) groups, respectively. Six of the 17 subjects who discontinued the training were not incorporated due to the lack of a will to participate. The same exclusion criteria as in 2014 was applied to the current study, but no one met them. All of these concerns were assessed using an interview sheet. The study protocol was approved by the Institutional Review Board of Osaka City University Graduate School of Medicine (approval no. 2719, approved on 9 January 2014) and was registered in the University Hospital Medical Information Network Clinical Trials Registry (UMIN000012781). Written informed consent was obtained from all participants after explanation of the study purpose. The study protocol also conformed to the ethical guidance of the 1975 Declaration of Helsinki.

### 2.2. Study Design

The present pilot study was designed as a nonrandomized, controlled study and was conducted between July 2014 and July 2016. Subjects in the EX group were engaged in a 2-year exercise program provided at Sumiyoshi Sports Center, a gymnasium located in Osaka City. Meanwhile, subjects in the NO group did not receive any exercise intervention. All subjects visited our research center in Osaka City University before and after the 2-year study period, and their cognitive function, body composition and motor ability were evaluated. 

### 2.3. Exercise Intervention

Subjects in the EX group received 1-h exercise training once a week for 2 years. All sessions were supervised by a trained instructor. A training session consisted of 15 min of mental gymnastics mainly made up of complicated finger movements, 25 min of resistance training, 10 min of aerobic exercise, and finally 10 min of systemic flexibility exercise in 8–10 poses. Resistance training gradually progressed from exercises like knee extensions or thigh raises in a sitting position to squats or back kicks in a standing position, or push-ups, or hip raises in a recumbent position on a mat. Aerobic exercises included stepping, simple walking, and zigzag walking with cones. DT training was incorporated into the sessions; concurrent cognitive tasks were performed during resistance training and aerobic exercises. For instance, one day, they conducted arithmetic tasks (subtraction of one digit) or played Shiritori, a Japanese word chain game in which one player has to say a word starting with the last character of the word given by the previous player during thigh raises. On the other day, they tried to switch the directions of trotting from forward to backward, or from left to right according to the pattern of whistling. In addition, they were tasked to retrieve as many names of animals as possible during stepping.

### 2.4. Demographic Data and Anthropometry

Demographic data including age, sex, and educational history were collected using interview sheets. Body mass index (BMI) was calculated as body weight × (height)^2^ and expressed in kilograms per square meter. Percentages of body fat and muscle mass of lower extremities were estimated by bioelectrical impedance analysis using a body composition analyzer (Nippon Shooter Ltd., Physion MD, Tokyo, Japan).

### 2.5. Cognitive Function

Cognitive assessments were performed before and after the intervention. In the EX group, the pre-intervention assessments were conducted at least 1 week before the first training session, and the post-intervention assessments at least 3 days after the last training session.

Cognitive performance was assessed using the Japanese version of the Modified Mini-Mental State (3MS) examination. This version was faithfully translated from the original version developed by Teng [15], and its usefulness has been properly validated [16]. The 3MS examination comprises 15 questions, which are categorized into eight domains, i.e., “registration and recall”, “long-term memory”, “orientation”, “attention”, “verbal fluency and understanding”, “word retrieval”, “visuospatial skills”, and “abstract meaning”. We graded each domain as well as the total scores (full marks = 100) of the 3MS examination.

The Trail-Making Test (TMT) was also performed to evaluate visual information-processing speed. We enabled easier and more systematic test processing using a dedicated device for TMT with a touch panel (Nounenreikei ATMT, elk Corp, Tokyo, Japan), in which a subject could pick the targeted circled number by touching it instead of drawing a line. The time taken in seconds to complete all 25 numbers was calculated automatically and used as the result.

All assessments of cognitive function were performed by a disinterested third person who did not know that the subject was in the EX group or NO group.

### 2.6. Quadriceps Isometric Muscle Strength and Motor Ability

Quadriceps muscle strength was assessed using the strain gauge dynamometer (ST-200S, MUL-TECH, Tokyo, Japan). Each subject performed two attempts on each leg, and the maximum value of all four trials was adopted.

Three physical performance tests were performed to evaluate motor ability, namely, maximal step length (MSL), Timed Up and Go (TUG) test, and single-leg standing. We measured the maximal time a subject could continue to stand on one leg with eyes open for up to 120 s. The best value from two trials was adopted.

### 2.7. Statistical Analysis

Data are presented as mean ± standard deviation. The comparison of baseline characteristics in both groups was performed by the unpaired *t*-test, except for the gender, medication, and medical history with the chi-square test. The effects of the exercise intervention on clinical parameters were examined by two-way repeated-measures analysis of variance; this procedure was repeated for both successive intervals within groups and between the NO and EX groups. The paired *t*-test was used in the case of a significant time (intervention) effect. In the case of significant trial (group) effect, subsequent comparisons were performed by the unpaired *t*-test. All statistical procedures were performed using SPSS Statistics (version 24.0, IBM, New York, NY, USA). *p*-values of < 0.05 were considered statistically significant. Ninety-five percent confidence intervals (CI) and F-values were calculated to estimate the strength of the association when the *p*-values for the group comparison were significant.

## 3. Results

### 3.1. Clinical Characteristics of the Subjects

Figure 1 shows the flowchart of the screening, baseline measurement, enrollment, intervention, and data analysis. No subject in both groups dropped out during the study period. We confirmed that the subjects in the EX group were engaged in > 90% of all sessions.

Table 1 shows the clinical characteristics of the subjects. All two male subjects were assigned to the NO group, while the EX group comprised only female subjects, although the difference between the groups was not statistically significant. At baseline, no differences in age, education, medication, and medical history were found between the groups. 

### 3.2. Effects of DT Training on Body Composition, Muscle Strength, and Motor Ability

Table 2 shows the changes in body composition, muscle strength, and motor ability after the intervention in both groups. At baseline, no differences in these parameters were found between the groups. BMI was decreased after the intervention only in the EX group without significant time × trial interaction.

Quadriceps muscle strength and MSL were significantly decreased during the 2-year study period similarly in both groups. The result of TUG test also deteriorated in both groups. Marked reduction in the maximal time of single-leg standing was found in the NO group. The post-intervention values of these measurements were not different between the groups.

### 3.3. Cognitive Function Outcome

Figure 2 shows the changes in the result of 3MS examination during the study period in both groups. At baseline, no differences in the total scores of the 3MS examination and scores of each domain of cognitive function were found between the groups, as well as in the results of the TMT.

A significant time × trial interaction was found on the total score of the 3MS examination (F (1, 17) = 7.682, *p* = 0.013), which meant a favorable effect of DT training on cognitive function. With regard to each component of 3MS examination, significant time × trial interactions were also found in the scores of two domains, namely, “attention” (F (1, 17) = 6.535, *p* = 0.020) and “verbal fluency and understanding” (F (1, 17) = 6.737, *p* = 0.019). The total score and the scores of “attention” and “verbal fluency and understanding” in the 3MS examination were decreased only in the NO group that withdrew from all exercise for the 2 years, whereas the subjects in the EX group successfully maintained those scores (“total score of the 3MS examination” NO group, 97.2 ± 2.6 to 91.6 ± 6.3, *p* = 0.009, 95% CI 1.75–9.52; EX group, 95.6 ± 4.4 to 96.3 ± 2.6, *p* = 0.559, 95% CI −3.64–2.14, “attention” NO group, 6.4 ± 1.2 to 5.0 ± 1.6, *p* = 0.023, 95% CI 0.23–2.50; EX group, 6.1 ± 1.6 to 6.5 ± 1.4, *p* = 0.351, 95% CI −1.26–0.51, “verbal fluency and understanding” NO group, 20.9 ± 0.3 to 20.1 ± 0.6, *p* = 0.004, 95% CI 0.29–1.16; EX group, 20.8 ± 0.3 to 20.8 ± 0.3, *p* = 1.000, 95% CI −0.45–0.45). As a consequence, the total score of the 3MS examination and the score in “verbal fluency and understanding” were significantly higher in the EX group than in the NO group after the intervention (NO group vs. EX group; 91.6 ± 6.3 vs. 96.3 ± 2.6, *p* = 0.044, 95% CI 0.16–9.32; 20.1 ± 0.6 vs. 20.8 ± 0.3, *p* = 0.006, 95% CI 0.23–1.16, respectively). No significant change was found in scores of other cognitive domains of the 3MS examination nor in the results of the TMT following the intervention in either group.

## 4. Discussion

The objective of the present pilot study was to investigate the effect of 2-year cognitive–motor DT training on cognitive functions and motor ability in healthy elderly people without marked cognitive impairment. The elderly subjects who were not engaged in any type of exercise for 2 years showed deterioration of 3MS examination result, especially in the attention domain. Meanwhile, participation in the DT training once a week could maintain almost all domains of cognitive function, as well as the total score of the 3MS examination. Although the findings need to be interpreted carefully because of the nonrandomized design, it seems that an exercise program composed of DT training is beneficial to prevent cognitive decline in healthy elderly people.

Various types of exercise are effective in decreasing the incidence of dementia and in delaying cognitive decline in patients with cognitive impairment. Previous studies suggested a greater effect of aerobic exercise, such as brisk walking or use of the stationary bicycle, on improving selective attention, memory, and processing speed in older adults with mild cognitive impairment [17,18]. Other studies have demonstrated that resistance training benefits the executive cognitive function, such as selective attention, memory, and conflict resolution, in elderly without cognitive impairment [19,20]. Therefore, a combination of aerobic and resistance training is often conducted in exercise programs such as in community activities that target elderly dwellers. Recently, DT training has attracted much attention for its potential benefit in terms of improving cognitive function of elderly people [14,21,22,23]. In DT training, two or more tasks are simultaneously performed, although the types of tasks included vary depending on the reports. In general, cognitive tasks, such as calculation or word retrieval, are often performed during aerobic or resistance training, such as walking or stepping. In addition, our 12-week intervention trial conducted in 2014 has demonstrated that DT training was more effective than single-task training in improving various cognitive domains [14]. Thus, evidence regarding the role of exercise as a non-pharmacological therapeutic strategy for cognitive impairment is now accumulating.

On the contrary, a small number of studies have examined the long-term outcome of aerobic, resistance, and balance training in modifying executive functions [17,24]. In particular, very few studies were conducted on the long-term effects of DT training on cognitive function. In the present study, the exercise program composed of DT training prevented deterioration in the results of the 3MS examination of elderly people, with mean age of 75 years, for 2 years. Because the 3MS examination is merely a device to screen the possibility of dementia, our findings do not immediately imply the direct effect of DT training on cerebral executive function. However, it does not seem that the maintenance of the score in the 3MS examination found in the EX group entirely depended on the fact that they repeated the cognitive tasks in the DT training. The training did not include saying a word or numbers backwards, as an example, which is part of the questions evaluating the domain of attention in the 3MS examination. A long-term perspective is needed for promoting health and independence of daily life for elderly people and for constructing sustainable social systems even in this unprecedented aged society. Therefore, the long-term outcome of DT training in the present study could cause a stir in the development of exercise programs that aim to maintain cognitive function of elderly people.

Moreover, increase in cerebral blood flow [25] and angiogenesis [26] by exercise is recognized to improve cognitive function. Ohsugi et al., reported that DT training significantly increased blood flow and the activity assessed by the quantity of oxygenated hemoglobin in the prefrontal cortex, the primary brain area that exerts executive function [22]. Therefore, the favorable effect of DT training on cognitive function maintenance may be, at least in part, because of increase in cerebral blood flow. Recently, as candidates, investigators suggest some neurotrophic factors that enhance cognitive function in relation to physical activity. Neurotrophic factors are proteins that promote neurogenesis and angiogenesis [25,26], resulting in increased hippocampal volume and improved memory function [27]. Higher serum levels of neurotrophic factors, such as brain-derived neurotrophic factor (BDNF) and insulin-like growth factor-1 (IGF-1), are associated with lower incidence of dementia and lower risk of progression in cognitive impairment in elderly people [27,28]. Furthermore, aerobic exercise is known to increase serum BDNF and consequently increase hippocampal volume [27]. Resistance training also increases serum and hippocampal IGF-1 levels in rats [29]. In addition, we detected significant increase in serum BDNF levels after the 12-week exercise intervention in our previous trial conducted in 2014 (data not shown). Therefore, BDNF can possibly contribute to the exercise-induced improvement in cognitive function in the trial. However, the superiority of DT training could not be explained by BDNF because the increase in serum BDNF levels by DT training was similar to that by single-task training. Further studies are needed to elucidate the molecular mechanisms of enhancing cognitive functions through exercise, which is specific to DT training.

In the present study, participation to the exercise program once a week could not prevent decline in the motor ability of elderly subjects. The American College of Sports Medicine recommends in their guideline of physical activity that healthy adults are engaged in aerobic exercise 3–5 days a week combined with resistance exercise for 2–3 days per week to improve and maintain physical fitness [30]. Currently, appropriate exercise frequency, intensity, and duration for elderly people with cognitive impairment have not been necessarily established. A systematic review concluded that it is essential for elderly people with cognitive impairment to engage in exercise for at least 45 min a day, 3 days per week for 12 weeks or more to improve physical function such as gait speed, functional mobility, balance, and activities of daily living [13]. Thus, considering these aspects, participation of elderly people with or without dementia in aerobic and resistance exercises 2–3 times a week to maintain muscle fitness is necessary.

The present pilot study has some limitations. First, a nonrandomized design with small sample size may have been insufficient to verify the difference between the NO and EX groups. Also, six of the 17 candidates who did not continue the training were not incorporated into the NO group, and this may consequently have resulted in study bias; Second, the daily physical activity levels in the EX group may have been greater than those in the NO group because of their higher awareness with respect to health and exercise, although we confirmed that the participants in the EX group had no special exercise habits except for our DT training. Furthermore, visiting to the sports club itself may result in an increase in physical activity in the EX group. This phenomenon may have influenced maintenance in the EX group; Third, we did not evaluate the socio-cultural factors, such as household income and occupation, which may have influenced to cognitive decline in the NO group [31]. Fourth, an imbalance in gender between the groups may have contributed to the greater decline of cognitive function in the NO group; all male subjects were assigned to the NO group. Some questions on word retrieval in the 3MS examination may be easy for females to score (e.g., list names of vegetables as many as possible). Finally, the decline in motor ability in the EX group during 2 years may result not from the low training frequency, but from the inadequate exercise intensity. Exercise intensity was not set according to the abilities of individuals because of the characteristics of our group exercise. Therefore, exercise intensity may not have been satisfactory for the subjects with higher muscle fitness.

## 5. Conclusions

In conclusion, the dual-task training seems to be beneficial in maintaining the broad domains of cognitive function in healthy elderly people for 2 years, although the findings need to be interpreted carefully because of the nonrandomized design. It is desirable that this low-cost, effective training is widely practiced by the population to withstand dementia. Further studies will elucidate the greater cognitive benefits by dual-task training than by single-task training in the long term follow-up.

Meanwhile, determining the ideal exercise prescription that targets physical performance and motor ability should be considered.

## Figures and Tables

**Figure 1 brainsci-08-00086-f001:**
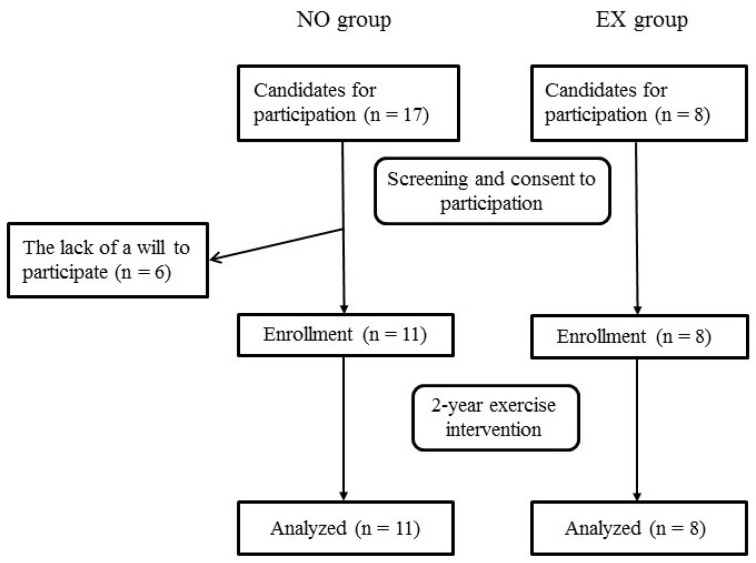
Flowchart of the screening, enrollment, intervention, and data analysis of the study. Abbreviations: NO, non-exercise; EX, exercise.

**Figure 2 brainsci-08-00086-f002:**
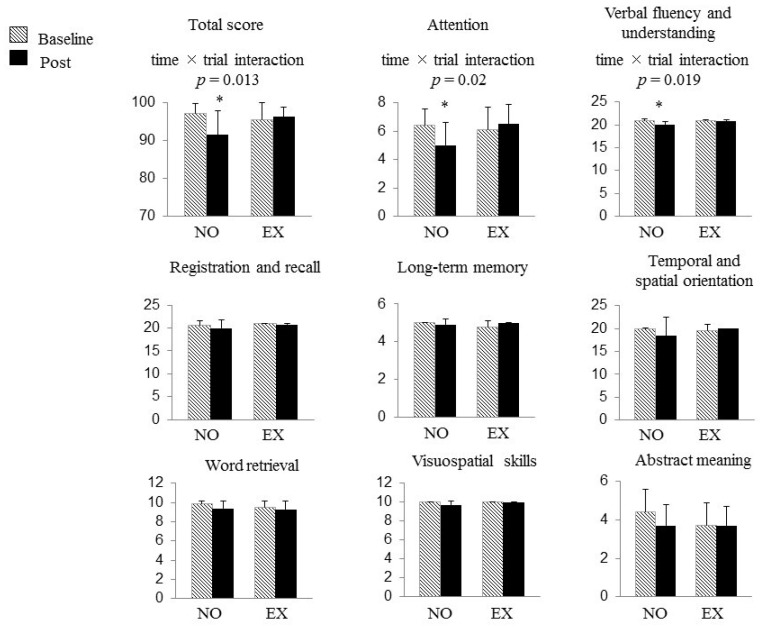
The changes in the results of the Modified Mini-Mental State (3MS) examination and Trail Making Test (TMT) during the study period in both groups. At baseline, no differences in the total scores of the 3MS examination, the scores of each domain of cognitive function, and the results of the TMT were found between the groups. The total score and the scores of “attention” and “verbal fluency and understanding” in the 3MS examination were decreased only in the NO group after 2 years, whereas the subjects in the EX group successfully maintained those scores. The horizontal bars represent means ± SD. *: *p* < 0.05 vs. baseline. Abbreviations: NO, non-exercise group; EX, exercise group.

**Table 1 brainsci-08-00086-t001:** Clinical characteristics at baseline.

	Overall (*n* = 19)	Non-Exercise Group (*n* = 11)	Exercise Group (*n* = 8)	*p*
Gender (*n*, %)				0.202
Male	2 (10.5%)	2 (18.2%)	0 (0%)	
Female	17 (89.5%)	9 (81.8%)	8 (100%)	
Age (years)	73.2 ± 3.5	71.9 ± 4.0	75.0 ± 1.5	0.056
Education (years)	12.3 ± 1.7	12.7 ± 1.9	11.6 ± 1.0	0.127
Medication (*n*, %)				0.633
Yes	9 (47.4%)	5 (45.5%)	4 (50.0%)	
No	10 (52.6%)	6 (54.5%)	4 (50.0%)	
Past history (*n*, %)				0.361
Yes	7 (36.8%)	5 (45.5%)	2 (25.0%)	
No	12 (63.2%)	6 (54.5%)	6 (75.0%)	
Knee osteoarthritis	2 (10.5%)	1 (9.1%)	1 (12.5%)	0.811
Cancer	3 (15.8%)	2 (18.2%)	1 (12.5%)	0.737
Others	2 (10.5%)	2 (18.2%)	0 (0%)	0.202
Present illness (*n*, %)				0.463
Yes	10 (52.6%)	5 (45.5%)	5 (62.5%)	
No	9 (47.4%)	6 (54.5%)	3 (37.5%)	
Hypertension	6 (31.6%)	2 (18.2%)	4 (50.0%)	0.141
Hyperlipidemia	4 (21.1%)	2 (18.2%)	2 (25.0%)	0.719
Others	4 (21.1%)	2 (18.2%)	2 (25.0%)	0.719

Age, Education: Values are presented as n or mean ± standard deviation (SD).

**Table 2 brainsci-08-00086-t002:** Blood pressure, body composition, muscular strength, and motor ability before and after the 2-year intervention.

		Non-Exercise Group (*n* = 11)		Exercise Group (*n* = 8)		*p*		
		Baseline	Post	Baseline	Post	Main Effect of Intervention	Main Effect of Group	Intervention × Group Interaction
SBP	(mmHg)	138 ± 17	144 ± 19	136 ± 17	135 ± 15	0.464	0.501	0.286
DBP	(mmHg)	79 ± 10	82 ± 10	82 ± 9	85 ± 9	0.267	0.499	0.918
BMI	(kg/m^2^)	22.5 ± 3.7	21.9 ± 4.1	22.4 ± 4.6	21.9 ± 5.0 *	0.006	0.968	0.889
Body fat	(%)	27.2 ± 6.4	26.6 ± 8.4	27.8 ± 6.5	27.2 ± 6.7	0.307	0.852	0.986
Leg muscle mass	(kg)	8.7 ± 2.4	8.6 ± 2.4	7.6 ± 1.1	7.7 ± 0.9	0.903	0.269	0.593
Quad. muscle strength	(kg)	30.5 ± 8.1	26.9 ± 8.7 *	27.4 ± 6.0	20.8 ± 4.1 *	<0.001	0.173	0.119
MSL	(cm)	109.5 ± 10.8	86.8 ± 10.0 *	103.5 ± 12.2	83.8 ± 10.1 *	<0.001	0.338	0.455
TUG	(sec)	5.28 ± 0.73	6.49 ± 0.71 *	5.61 ± 0.67	6.67 ± 0.99 *	<0.001	0.443	0.678
Single-leg standing	(sec)	87.4 ± 44.1	57.4 ± 45.6 *	87.9 ± 42.1	71.6 ± 48.9	0.008	0.711	0.388

All values are presented as *n* or mean ± SD. * *p* < 0.05 within the group. Abbreviations: SBP, systolic blood pressure; DBP, diastolic blood pressure; BMI, body mass index; MSL, maximal step length; TUG, Timed Up and Go test.

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
