# Peer review of "Effects of 2-Year Cognitive–Motor Dual-Task Training on Cognitive Function and Motor Ability in Healthy Elderly People: A Pilot Study"

_brainsci, 2018, doi:10.3390/brainsci8050086_

Round 1

Reviewer 1 Report

The research project of a 2-year simultaneous motor and cognitive tasks training program is very interesting.

It has the limitation that the sample size is extremely small and the selection to the intervention group is voluntary (not randomized or with matched controls). Therefore, the findings have to be interpreted rather as a pilot study.

I have two questions:

What diseases (diabetes, hypertension, stroke, etc.) did the participants have and how did it affect the results?

What cognitive tasks did the training comprise (describe in detail because this is the relevant finding)?

With respect to the statistical methods, I would recommend mixed-models (random and fixed-effects) or at least a repeated measures ANOVA. It would be great if you could adjust the analyses for age, gender and other covariates.

Author Response

We are grateful to Reviewer 1’s favorable general comments. Here, we would like to address point-by-point to Reviewer 1’s comments or questions.

The present study was conducted as a nonrandomized design with small sample size. As Reviewer 1 pointed out, the results of the study have to be treated as a pilot study. Therefore, we defined it in the title and made some revisions as follows:

P1  line 4  

P2  line 52, line 79

P7  line 208

P8  line 278 – 281

As we have described in the 2.1. Participants paragraph in the Materials and Methods section of the original manuscript, the applicants who had a history of ischemic heart disease, chronic heart failure, severe hypertension, diabetes, marked dementia, or a neuropsychiatric disorder were excluded from the study because they could affect the improvement of cognitive function by the training. One applicant with marked dementia was excluded in our previous study in 2014, and no one met the exclusion criteria in the current study. To clarify this process of exclusion, we revised the description in the 2.1. Participants paragraph and Table1 as follows:

P2  line 58 – 64, line 69-72

To response to Reviewer 1’s natural question regarding the detail of the cognitive tasks, we added some descriptions to the 2.3. Exercise intervention paragraph in the Materials and Methods section. Our revisions are follows:

P3  line 94, line 97 - 99          

In regard to the statistical methods, we have already used ANOVA with repeated measures to examine the effects of exercise intervention on clinical parameters, and the issue was described in the 2.7. Statistical analysis paragraph in the Materials and Methods section of the original manuscript. However, to make this issue easy to understand, we added the p-values for time × trial interaction to Figure 2. It would also have been good if we could adjust the analysis for age, gender and other covariates. However, we could not do so because of small sample size. We will try to increase the number of samples in the future study and examine them.

Reviewer 2 Report

This manuscript examines the impact of a dual task (motor and cognitive) training program in healthy elderly for 2 years.  The study sample is one of convenience (they recruited individuals from a previous study), and the group assignment was not random (they asked those from the previous study to continue with more DT, and those that said no were the control group).  Thus, while this study is a preliminary look at the long-term impact of DT on cognitive and physical exercise, the conclusions that can be drawn are highly limited.  This fact should be highlighted throughout the text, including in their limitations section, and the strong wording of cognitive benefit should be toned down and caution in interpretation added to the first paragraph of the discussion and the conclusions.

Additionally, there is reference sporadically throughout about the impact on executive functions.  They do not clearly measure executive functions well in this study.  Rather, they have a global screening measure of cognition (3MS) and what appears from the description to be Trail Making Test A done on a computer.  Neither of these are true executive function tests, and wording should be changed to reflect that what they really saw was improvement in attention and verbal fluency, as measured by the 3MS.  Still, I have some concerns about even that conclusion.  Namely, were any of the cognitive exercises similar to the attention task on the 3MS?  Were they practicing language fluency (like in the game of Shiritori they describe)?  Thus, are they only practicing the task itself rather than having a true impact on attention and verbal fluency?  If so, this significantly limits the meaningfulness of this study. Further, I am surprised that with such a small difference in mean change in the verbal fluency groups (0.8 in the NO group and no change in the EX group was significant in this small of a sample).  Can the authors confirm this and the statistics used?

Additional concerns that need addressed include:

What was the reason for the 6 screen failures for enrollment (Figure 1)?  And, is there reason for concern for more bias in the samples?

Were there any cognitive differences at baseline?

The difference in gender amongst groups needs to be highlighted sooner (e.g., early in the results) than waiting till the limitations section.

How was BMI significantly different?  The numbers look essentially the same in Table 1. 

Author Response

We appreciate Reviewer 2’s time and effort to review our manuscript. Here, we would like to address point-by-point to Reviewer 2’s comments or questions.

Major points

As Reviewer 2 pointed out, we have to interpret our results carefully due to a nonrandomized design with small number of the subjects recruited from the participants of our previous study. According to the comment, now we treat our findings as a pilot study, and toned down the representation in the Discussion and the Conclusions sections. Revisions are as follows:

P1  line 4, line 21, line 33

P2  line 52 – 53, line 79

P7  line 208 – 209, line 213 – 214

P8  line 265, 278 – 281, line 298 - 301 

While the 3MS and Trail Making Test are globally accepted tools to screen cognitive impairment of suspected elderly persons of dementia, neither of them can directly examine the executive functions. Therefore, according to Reviewer 2’s comment, we have to interpret our results carefully as mentioned above, and now toned down the representation in the Discussion and the Conclusion sections. On the other hand, we demonstrated the effect of dual-task training on maintenance of the score in the 3MS, particularly in the “attention” domain. In the 3MS, attention is evaluated by the questions such as saying a word or numbers backwards. It does not seem that the maintenance of the total score and the score of “attention” in the 3MS found in the EX group was entirely resulted from the practice effect of the 2-year training because the cognitive tasks did not include saying a word or numbers backwards. To clarify this issue, we revised the Discussion section as follows:   

P7  line 236 – 242

3.  Regarding the difference in the changes in the score of verbal fluency between the groups, no statistical mistakes were found after the reconfirmation. Almost all of the subjects have got full marks in the questions to assess verbal fluency, and therefore a slight deduction in the scores might have resulted in a statistical significance.

Minor revisions

1. Six of the 17 candidates who did not continue the training were not incorporated into the NO group due to the lack of a will to participate. To clarify this issue, we revised the description in the 2.1. Participants paragraph and Figure 1. As Reviewer 2 pointed out, the issue may also have resulted in sampling bias. Therefore, we added the description in the Discussion section. Revisions are as follows:

P2  line 69 – 70

P8  line 279 – 281 

2. At baseline, no differences in the total scores of the 3MS examination and scores of each domain of cognitive function were found between the groups. To clarify these results, we revised the description in the 3.3. Cognitive function outcome paragraph in the Results section, and that in the legend of Figure 2. Revisions are as follows:

P6  line 182, line 177 – 178 

3. According to the recommendation by the Reviewer 2, we refered to the difference in gender between the groups in the 3.1. Clinical characteristics of the subjects paragraph in the Results section of the revised manuscript. Revisions are as follows:

P4  line 153 – 155

4. Regarding BMI, it also tended to decrease after the intervention in the NO group, although the change did not reach statistical significance.

(NO group, 22.5 ± 3.7 to 21.9 ± 4.1, p = 0.053; EX group, 22.4 ± 4.6 to 21.9 ± 5.0, p = 0.035).

Reviewer 3 Report

Introduction: Physical activity is different from motor training. Consider just using a single term. In addition, there is little or no mention of cognitive training literature in the introduction. 

Page 9,  2.1. Participants:  

Need to add some background of the original 25 participants, how they were recruited, demographic characteristics, etc.? Small sample size is a big concern. Also, are those 6 who discontinued the training and did not consent to participate in the present study different from those 11 who did participate in the current study?  

For “Subjects who had a history of ischemic heart disease, chronic 64 heart failure, severe hypertension, diabetes, or neuropsychiatric disorder were excluded from the 65 study.”: how many of such participants were excluded? At which stage were they excluded?

It seems it was not a random selection into EX or NO group. Is selection bias a concern? Were those who voluntarily participated in the EX program more health conscious people? Or were they in bad health condition so they were more motivated to do exercise?

Page 3, 2.5. Cognitive function : When (before, during, or after ) was the cognitive performance assessed?  

Page 10: “At baseline, no differences in age, 137 education, medication, body composition, muscle strength, and motor ability were found between 138 the groups.” With the small sample size, how likely would the authors be able to see a significant difference? What is the power for detecting any difference?

Table 1, what are the p-values referring to?

The dual-task training design, without further single-training groups, of the study made it hard to differentiate whether the cognitive benefits are due to physical exercise or cognitive training. The less decline in cognition among the EX group might just be due to learning effects.

How the study make sure those in NO group did not exercise by themselves?

Author Response

We appreciate Reviewer 3’s time and effort to review our manuscript. Here, we would like to address point-by-point to Reviewer 3’s comments or questions.

1. In the original manuscript, we used “physical activity” in a meaning that someone has exercise habit (not intervention). As Reviewer 3 pointed out, the coexistence of “physical activity” and “motor training” is confusing. Now we have deleted the sentence including “physical activity” from the Introduction section, and unified the word into “exercise” throughout the manuscript.

  Additionally, according to Reviewer 3’s comment, we cited some litereatures about the cognitive training in the Introduction section as follows:

  P 1   line 43 – 44

  P 2   line 45 – 46

2. To address the concerns of Reviewer 3 regarding the participants, we added the detail to the 2.1. Participants paragraph in the Materials and Methods section. The original 25 participants were recruited from community dwellers in Sumiyoshi-ku, Osaka City, Japan by advertising in a local magazine. One applicant with marked dementia was excluded in our previous study in 2014. In the current study, no one met the same exclusion criteria as in 2014 at the enrollment. Revisions in the 2.1. Participants paragraph are as follows:

P2   line 58 – 64, line 69 - 72

3.  As Reviewer 3 pointed out, we have to interpret our results carefully due to a nonrandomized design, in which the subjects assigned to the EX group may have had the greater health consciousness, and that may have resulted in the selection bias. Now we treat our findings as a pilot study, and refered to the issue in the Discussion section.

       P1  line 4    

P2  line 52, line 79

P7  line 208

P8  line 278 – 281

4. Cognitive assessments were performed before and after the intervention. In the EX group, post-intervention assessments were conducted at least 3 days after the last training session. According to the comment by Reviewer 3, we revised the description in the 2.5. Cognitive function paragraph in the Materials and Methods section as follows:

P3  line 108 – 109

5.  As Reviewer 3 pointed out, the small number of participants may have been insufficient to verify the difference between the NO and EX groups. Therefore, we refered to the issue in the Discussion section. We also added ninety-five percent of confidence intervals (95 % CI) and F-values to the results as the power for detecting difference. Revisions are as follows:

P4  line 141 – 143

P6  line 189 – 190, line 192 – 193, line 197 – 201, line 204

P8  line 278 – 281 

6.  The p-values in Table 2 (Table 1 in the original manuscript) refers to the main effects of intervention and group, and intervention × group interaction by ANOVA with repeated measures, respectively. To make them easy to understand, we revised Table 2.

7.  In the duration of the present study, we could not unfortunately open the training course for signal-task training. As Reviewer 3 pointed out, it would have been good if we could set the single-task training group as the control to elucidate the superiority of the dual-task training compared to the single task training.

We refered to this issue in the Conclusions section as follows:

P8  line 298 – 301

8. We demonstrated the effect of dual-task training on maintenance of the score in the 3MS, particularly in the “attention” domain. In the 3MS, attention is evaluated by the questions such as saying a word or numbers backwards. It does not seem that the maintenance of the total score and the score of “attention” in the 3MS found in the EX group was entirely resulted from the learning effect of the 2-year training because the cognitive tasks did not include saying a word or numbers backwards. To clarify this issue, we revised the Discussion section as follows:

P7  line 236 – 242

9. All of the concerns about the applicants, such as exercise habits, past history, present illness, and medication were assessed using an interview sheet. According to the assessment, we ascertained that the subjects in the NO group were not engaged in any type of exercise for more than 1 hour per week. To clarify this issue, we added the description in the 2.1. Participants paragraph in the Materials and Methods section as follows:

 P2  line 66 – 67, line 71 – 72

Round 2

Reviewer 1 Report

The authors have fully addressed my comments.

Reviewer 2 Report

The abstract still needs to be toned down from suggesting that the training "is beneficial" to "may be beneficial" but requires further study.

In regards to the edits added on p. 8 line 265.  This addition is hard to read and unclear.  It isn't '"practice effects" per say that is of concern, as "practice effects" usually refers to the impact of repeatedly taking the test.  The concern is that participants were trained to the test task itself.  I believe this just needs re-worded and better explained, as it seems the authors are suggesting that this did not happen.

Author Response

Once again we appreciate Reviewer 2’s time and effort to re-review our manuscript. Here, we would like to address point-by-point to Reviewer 2’s comments or questions.

1. According to Reviewer 2’s suggestion, we revised the Abstract section as follows:

P1  line 30-33

        These results suggest that participating in exercise program comprising DT training for 2 years may be         beneficial to maintain the broad domains of cognitive function in healthy  elderly people, although the         further verifications are needed.

2. We used the “practice effects” in a meaning, as Reviewer 2 meant, of repeating the cognitive tasks comprising the 3MS questions in the training. As we mentioned in the former response, we don’t think that the maintenance of the score in the 3MS found in the EX group entirely depended on that they repeated the cognitive tasks in the training. It is because the cognitive tasks did not include saying a word or numbers backwards which comprising the questions evaluating the domain of attention in the 3MS examination. However, as Reviewer 2 pointed out, our revision might have been unclear. Now we revised the corresponding descriptions again as follows:  

P7  line 240 – 246

       However, it does not seem that the maintenance of the score in the 3MS examination found in the EX         group entirely depended on that they repeated the cognitive tasks in the DT  training. It is because the         training did not include saying a word or numbers backwards, as a example, which comprising the          questions evaluating the domain of attention in the 3MS examination.

Reviewer 3 Report

The authors have responded to most of my previous questions. However, the  timing of pre-intervention cognitive assessment needs further clarification. How long ( how many days?) before the intervention? Also, the learning effects from the intervention on the outcomes needs to be discussed. In addition, if the authors decided to use “exercise”, should the paper title also be changed accordingly? It looks “cognitive training” component of the intervention is minimal, while the title implicates it is the key part of the intervention. 

Author Response

Once again we appreciate Reviewer 3’s time and effort to re-review our manuscript. Here, we would like to address point-by-point to Reviewer 2’s comments or questions.

Regarding the timing of the cognitive assessments before the intervention, it was at least 1 week before the first training session. We added the detail as follows:

P3  line 109 – 111

In the EX group, the pre-intervention assessments were conducted at least 1 week before the first training session, and the post-intervention assessments at least 3 days after the last training session. 

Regarding the learning effects, as we mentioned in the former response, we don’t think that the maintenance of the score in the 3MS found in the EX group entirely depended on that they repeated the cognitive tasks in the training. It is because the cognitive tasks did not include saying a word or numbers backwards which comprising the questions evaluating the domain of attention in the 3MS examination. However, our revision according to the former comment by Reviewer 3 might have been unclear. Now we revised the corresponding descriptions again as follows:  

P7  line 240 – 246

However, it does not seem that the maintenance of the score in the 3MS examination found      in the EX group entirely depended on that they repeated the cognitive tasks in the DT  training. It is because the training did not include saying a word or numbers backwards, as a example, which comprising the questions evaluating the domain of attention in the 3MS examination.

As we mentioned in the former response, although we were misunderstood, we deleted the “physical activity” representing having a habit of exercise at a particular time, and unified the words into the “exercise” or the “training” which means exercise intervention instead. We thought that cognitive training was even significant in the maintenance of cognitive function as motor training, and that they have synergistic effects on cognitive function as simultaneously conducted. Therefore, we would like to leave the manuscript title unchanged.